# Discovery of New 2-Phenylamino-3-acyl-1,4-naphthoquinones as Inhibitors of Cancer Cells Proliferation: Searching for Intra-Cellular Targets Playing a Role in Cancer Cells Survival

**DOI:** 10.3390/molecules28114323

**Published:** 2023-05-24

**Authors:** Julio Benites, Jaime A. Valderrama, Álvaro Contreras, Cinthya Enríquez, Ricardo Pino-Rios, Osvaldo Yáñez, Pedro Buc Calderon

**Affiliations:** 1Química y Farmacia, Facultad de Ciencias de la Salud, Universidad Arturo Prat, Casilla 121, Iquique 1100000, Chile; juliob@unap.cl (J.B.); jaimeadolfov@gmail.com (J.A.V.); alvaro.contreras92@gmail.com (Á.C.); fioutuanehs@gmail.com (C.E.); rpinorios@unap.cl (R.P.-R.); 2Departamento de Química Orgánica, Facultad de Química y de Farmacia, Pontificia Universidad Católica de Chile, Avenida Vicuña Mackenna 4860, Santiago 7820436, Chile; 3Núcleo de Investigación en Data Science, Facultad de Ingeniería y Negocios, Universidad de las Américas, Santiago 7500000, Chile; oyanez@udla.cl; 4Research Group in Metabolism and Nutrition, Louvain Drug Research Institute, Université Catholique de Louvain, 73 Avenue E. Mounier, 1200 Brussels, Belgium

**Keywords:** cancer cells, quinones, mTOR, antiproliferative activity, molecular descriptors, molecular docking

## Abstract

A series of 2-phenylamino-3-acyl-1,4-naphtoquinones were evaluated regarding their in vitro antiproliferative activities using DU-145, MCF-7 and T24 cancer cells. Such activities were discussed in terms of molecular descriptors such as half-wave potentials, hydrophobicity and molar refractivity. Compounds **4** and **11** displayed the highest antiproliferative activity against the three cancer cells and were therefore further investigated. The in silico prediction of drug likeness, using pkCSM and SwissADME explorer online, shows that compound **11** is a suitable lead molecule to be developed. Moreover, the expressions of key genes were studied in DU-145 cancer cells. They include genes involved in apoptosis (*Bcl-2*), tumor metabolism regulation (*mTOR*), redox homeostasis (*GSR*), cell cycle regulation (*CDC25A*), cell cycle progression (*TP53*), epigenetic (*HDAC4*), cell-cell communication (*CCN2*) and inflammatory pathways (*TNF*). Compound **11** displays an interesting profile because among these genes, *mTOR* was significantly less expressed as compared to control conditions. Molecular docking shows that compound **11** has good affinity with mTOR, unraveling a potential inhibitory effect on this protein. Due to the key role of mTOR on tumor metabolism, we suggest that impaired DU-145 cells proliferation by compound **11** is caused by a reduced *mTOR* expression (less mTOR protein) and inhibitory activity on mTOR protein.

## 1. Introduction

Quinones are ubiquitous in nature and their scaffolds are present in many drugs such as anthracyclines, daunorubicin, doxorubicin, mitomycin, mitoxantrones and saintopin (Figure 1). Some of them are used clinically in the therapy of solid cancers. The cytotoxicity of these quinones is typically caused by inhibition of DNA topoisomerase-II and/or the formation of reactive oxygen species (ROS), generated during quinone redox cycling [1]. The most important and widely distributed chemical class in the quinone family is that of 1,4-naphthoquinones. Their biological activities, particularly against cancer cells, have motivated many studies focusing on the development of novel antitumor agents based on the 1,4-naphthoquinone array [2,3,4,5]. In line with the necessity of alternatives drugs in cancer therapy, quinones are important tools in the search for intracellular targets that play a role in cancer cell survival. 

Among the diverse reported synthetic 1,4-naphthoquinone-containing analogues, 2-phenylamino-1,4-naphthoquinone **I** and their phenylamino-substituted derivatives, such as compounds **II** and **III** (Figure 2), have been investigated for their anticancer properties [6,7]. The synthetic flexibility towards 2-arylamino-1,4-naphthoquinones and the redox capability of the 1,4-naphthoquinone scaffold, determined by the magnitude of the donor effect of the arylamino substituents, have contributed, in part, to establishing proofs of the probable biological mechanism and targets involved in the anticancer effects. Note that these mechanisms and targets suggest that they participate in the cell redox cycle and act as precursors of ROS, which leads to oxidative stress [8].

We have previously shown the antiproliferative properties of a class of analogues of 2-arylaminonaphthoquinones containing an electron-acceptor acyl substituent located at the 3-position of the quinone scaffold, named 2-arylamino-3-acyl-1,4-naphthoquinone [9,10]. The screening of the members of the series, such as compounds **IV**–**VI** (Figure 2), express in vitro antiproliferative activity against non-tumor fibroblasts and a variety of cancer cell lines [9]. Additional studies on 2-arylamino-3-acyl-1,4-naphthoquinones in the MCF-7 human breast cancer cell line and in male Ehrlich tumor-bearing Balb/c mice demonstrated that compound **V** displays a high cytotoxicity against MCF-7 with IC_50_ values of 1.5 μM [10].

However, cancer cells display much higher ROS levels than normal cells due to dysfunctional mitochondria, oncogene activation and antioxidant imbalance [11,12]; for instance, ROS inactivate PTEN facilitating PI3K, Akt/mTOR signaling, which ultimately leads to tumor progression [13]. Two decades ago, Vafa et al. [14] reported that increased ROS levels activate c-Myc in a HIF1-α-dependent way, resulting in tumor proliferation and DNA damage. Since then, several studies led to the conclusion that loss of redox homeostasis is a hallmark of cancer cells [12,15,16,17], likely due to molecular interactions of ROS molecules with specific targets in redox signaling pathways. In line with previous work, we support the hypothesis that a pro-oxidant treatment significantly contributes to the elimination of cancer cells via the induction of an oxidative stress leading to different manners of cell demise [18,19,20,21,22]. Within this framework, we have synthesized several quinone compounds and their biological activities have been assessed on a variety of human cancer cells [23,24,25,26,27,28,29]. In short, we have shown that quinones are important tools in the search for intracellular targets that play a role in cancer cell survival. Their ability to interact with different intracellular targets and their synergistic effect with other anticancer treatments make them attractive therapeutic targets for cancer therapy.

The aim of the work was to investigate the in vitro cytotoxicity of a newly designed series of 2-phenylamino-3-acyl-1,4-napththoquinones, using three human cancer cell lines (DU-145, MCF-7 and T24) and healthy non tumor HEK-293 cells. The antiproliferative activities were examined in terms of molecular descriptors (half-wave potentials, hydrophobicity and molar refractivity) as well as by in silico prediction of drug likeness, using pkCSM and SwissADME explorer online. In addition, gene expression and molecular docking studies were performed to explore a potential association between antiproliferative activities and expression of some representative genes of major cellular pathways playing a role in carcinogenesis and cancer cells survival. These pathways include apoptosis, redox homeostasis, tumor metabolism regulation, cell cycle, epigenetic, cell-cell crosstalk and inflammation. In sum, our experimental design allows us to assess the influence of the stereoelectronic factors involved in biological mechanisms subtending the potential antiproliferative activity of the members of the new 2-phenylamino-3-acyl-1,4-naphthoquinones series.

## 2. Results and Discussion

### 2.1. Anti-Proliferative Evaluation

The structures of compounds **1**–**14** were designed to evaluate how their antiproliferative activities are influenced by the stereoelectronic effects of diverse electron-withdrawing acyl groups located at the 3-position of the 2-phenylamino-1,4-naphthoquinone core. To this end, compounds were tested against three human cell lines and the results are summarized in Table 1 The biological assays included the non-tumorigenic HEK-293 (embryonic kidney cells) and three human-derived cancer cell lines, namely, prostate (DU-145), breast (MCF-7) and bladder (T24). The antiproliferative activity of quinones was evaluated through their IC_50_ values, expressed in µM. Doxorubicin, a well-known anticancer drug, was included as a reference compound.

### 2.2. Physicochemical Descriptors

The selection of electron-withdrawing acyl groups (R-C=O) in the synthesized compounds was made to cover significant stereoelectronic structural differences among the members of the series. Therefore, the structure–activity relationships were examined regarding the following structural features of the ligands: (**a**) alkyl (compounds **1**–**3**); (**b**) phenyl (compounds **4**–**10)** and (**c**) heteroaryl (compounds **11**–**14**).

In addition, three standard molecular descriptors commonly used in structure–activity relationships of cytotoxic 1,4-naphthoquinones [1,2,3,4,30,31] were evaluated: half-wave potential (**E^I^**_½_, expressed in mV), hydrophobicity (ClogP) and steric effect as molar refractivity (CMR, expressed in cm^3^/mol). These parameters are key in order to delineate a large number of receptor–ligand interactions that are crucial to biological processes [1]. To this end, the previously mentioned physicochemical descriptors were acquired to obtain qualitative information on their putative correlation with the observed antiproliferative activities of the 2-phenylamino-3-acyl-1,4-naphthoquinone **1**–**14** (Table 2).

According to the data in Table 1 and Table 2, compounds **1**–**3**, having the alkyl-C=O ligands, show weak cytotoxic activity for **1** (IC_50_: 60.89 to 86.69 μM), while **2** and **3** were almost devoid of activity. Among these members, the most active compound **1** exhibited the lower values of lipophilicity (Clog P = 2.72) and molar refractivity (CMR = 9.96). In the case of the aryl-C=O and heteroaryl-C=O ligands, interesting antiproliferative activities in the IC_50_ range: 0.82 to 21.66 μM were observed. Inspection of structural features vs. antiproliferative activities of the aryl-C=O group members (**4**–**10**), led us to conclude that compounds **4** and **11**—the most active compounds among the aryl-C=O ligands and heteroaryl-C=O ligands, respectively—exhibited the lower CMR values (10.15 and 9.36).

Comparison of the ClogP vs. IC_50_ values of the members of aryl-C=O ligand reveals that lipophilicity does not influence the cytotoxic activity of their members **4**–**10**. Similarly, comparing the ClogP vs. IC_50_ values of the members of heteroaryl-C=O ligands, in particular that of **11** (ClogP = 1.24) and **14** (ClogP = 1.17)**,** i.e., the most against the less active members of this group, suggests that lipophilicity does not capture the variability of the biological activity. 

Inspection of the half-wave potentials of the members **1**–**14** of the series, located in the range −800 to −596 mV, reveals significant stereoelectronic effects for the members of the aryl- and heteroaryl-CO ligands **4**–**14**. Compounds **4** and **11** appear as the most active members of these groups, showing one-electron reduction capability in terms of their half-wave potentials at −695 mV and −578 mV, respectively.

Notably, the most bioactive members of the alkyl, aryl and heteroaryl-CO groups of the series exhibited the lower CMR values (9.96, 10.15 and 9.36). Based on these results, we can infer that molar refractivity could be a valuable parameter for the design of new members of the series endowed with cytotoxic activity.

Considering the mean selective index displayed by **4** (MSI: 6.2) and **8** (MSI: 4.7), we selected the former to be included in further gene expression and molecular docking studies. Regarding members of the group having the heteroaryl-C=O ligands, it is evident that due to structural analogies of the O,N,S-heterocycles involved into their structures, no significant differences among the CMR parameter were observed. Nevertheless, among this group, bioisosters **11**/**12** exhibited higher activity than the pair **13**/**14**, likely due to redox capability of **11**/**12** pair having less negative potential (−578 and −552 mV) than the **13**/**14** pair (−635 and −685 mV).

Complementary studies to those resulting from the SAR analysis were conducted to obtain some insights regarding the molecular mechanism involved in the in vitro antiproliferative evaluation of bioactive quinones. Among all the tested compounds **1**–**14**, compound **11**, displaying good antiproliferative activity, high hydrophilic character and low activity against healthy non-tumorigenic HEK-293 cells, was selected as a promising potential anti-cancer molecule. Therefore, compound **11** and its carbocyclic analogue **4** were included in the next studies (representative dose–response curves for compounds **4** and **11** are reported in the Appendix A. First, we investigated their physicochemical, pharmacokinetic and drug likeness properties. Second, we focused on gene expression and molecular docking studies. 

### 2.3. pkCSM and SwissADME

Drug development involves assessment for physicochemical properties, pharmacokinetics, drug likeness and medicinal chemistry friendliness; in that context, computer models constitute valid alternatives to experiments. Both pkCSM and SwissADME explorer online were used for in silico prediction of drug likeness of the synthesized compounds **4** and **11** based on various molecular descriptors and the results are depicted in Table 3.

The results obtained from ADMET analysis and depicted in Table 3 revealed that the structures **4** and **11** had a molecular weight smaller than 500 g/mol, which is important for penetrability [32]. Both molecules show Caco-2 permeability values below 1.00, as well as high intestinal absorption (94.3%), suggesting that they would be absorbed in the small intestine [33]. The transdermal efficacy as illustrated by skin permeability of compounds **4** and **11** was from −2.774 and −2.798 cm/hour, which mean that they will penetrate the skin properly. Note that molecules will penetrate the skin with difficulties if the logKp value is greater than −2.5 cm/hour [34]. Circulation in blood plasma (VDss) is acceptable for compounds with values higher than −0.15. Penetration via the blood–brain barrier (BBB) is an important parameter for reducing side effects and toxicity. Note that compounds **4** and **11** have log BB < 0.3, and thus would be able to penetrate the brain [35]. None of the compounds appeared to be CYP2D6 inhibitors, but they inhibited CYP3A4, a potential interference with CYP450 biotransformation reactions. Excretion parameters are illustrated as total clearance. They showed that compounds have positive values, indicating rapid excretion. In addition, the adverse interactions of both compounds with the organic cation transport 2 (OCT2) showed no potential contraindication. Finally, while both compounds did not violate Lipinski’s Rule of Five, compound **4**, but not compound **11**, shows hepatotoxicity. In summary, it seems that both compounds **4** and **11** are drug likeness structures allowing a further drug development, but they differ in terms of liver toxicity [36]. Additionally, the Brenk filter analysis [37] indicated molecular fragments that may be potentially metabolically unstable, toxic or responsible for poor pharmacokinetic behavior. Compounds **4** and **11** are Michael acceptors, with one carbonyl group belonging to the naphthoquinone skeleton. In contrast, the other carbonyl group is part of the acyl group. Although this is an alert to be taken into account, both carbonyls belong to different fragments. The full reports of SwissADME and pkCSM parameters are reported in Appendix A.

### 2.4. mRNA Expression Evaluation

Since neither a particular sensitivity nor a resistance against quinones were observed between the three cell lines utilized for antiproliferative assays, a follow-up study was conducted by using the DU-145 prostate cancer cell line. The selection of these cancer cells was made based on practical in house motives (i.e., best survival levels, growth rapidity, easy manipulation, etc.). Subsequently, the effects of compounds **4** and **11** were further explored looking for changes in gene expression levels in DU-145 cells and the mRNA levels were analyzed by RT-PCR and normalized to *B2M* levels. 

Table 4 shows the changes in the expression of different genes after 24 h incubation of DU-145 cells in the absence or presence of compounds **4** and **11**. As explained previously, these genes are representative of major pathways involved in carcinogenesis and cancer cell survival. They included genes involved in apoptosis regulation (*Bcl-2*), in kinases cascades regulating tumor metabolism (*mTOR*), in redox homeostasis and protection against oxidative stress (*GSR*), in regulation of cell cycle (*CDC25A*), in tumor suppression and cell cycle progression (*TP53*), in epigenetic and transcriptional regulation (*HDAC4*), in cell-cell communication (*CCN2*) and in inflammatory pathways (*TNF*).

In this assessment, only compound **11** displays an interesting profile; indeed, two genes were less expressed as compared with control conditions, namely, *Bcl-2* and *mTOR*. The former one has an anti-apoptosis function [38,39,40] and its depressed levels may be associated with a smaller cellular proliferation [41], but the inhibitory effect induced by compound **11** was not statistically significant as compared with control conditions (*B2M* levels). 

Interestingly, the effect of compound **11** on the second one, *mTOR*, is not only statistically significant but is also biologically relevant. Indeed, mTOR regulates different cellular processes such as cell growth, cell proliferation, cell motility, cell survival, protein synthesis, autophagy and transcription [42,43]. Furthermore, the activity of mTOR was found to be dysregulated in many types of cancer cells likely caused by mutations in tumor suppressor PTEN gene [44] and an increased activity of PI3K or Akt [45,46,47]. Consequently, the mTOR signaling pathway, which is often activated in tumors, significantly contributes to the initiation and development of cancer cells and it plays an important role in their metabolism [48,49,50]. Therefore, decreased levels in the expression of *mTOR* gene induced by compound **11** may be correlated with its antiproliferative effect. Such inhibition of *mTOR* expression (and the prediction of a decreased amount of mTOR protein) is relevant because the mTOR signaling pathway is dysregulated by increased activity of PI3K or Akt [49,50], and we have previously reported that a similar family of quinones, synthesized in our laboratory, has an inhibitory effect on Akt [27].

Since mTOR, a master protein regulating cancer cell metabolism and proliferation, may be impaired at two different levels, gene expression and protein activity, and given the observance of a positive correlation between mRNA and protein expression levels, we next explored the potential interactions of quinones with the gene products (proteins) that were analyzed in DU-147 cancer cells. 

### 2.5. Molecular Docking

Molecular docking simulations were performed to study the binding pattern of compounds **4** and **11** in the active sites of mTOR, Bcl-2, GSSG reductase, HDAC4, TNF-α, CDC25A and B2M proteins as shown in Table 5.

According to the results shown in Table 5, we observed that the interaction of compounds **4** and **11** with the selected proteins can be classified into three categories. Firstly, the best interactions are observed with the group formed by TNF-α, GSR and HDAC4, reaching values of −9.1 to −8.2 kcal·mol^−1^, with compound **4** presenting a slightly better interaction than compound **11**. We note that both quinones did not modify the expression levels corresponding to the genes encoding these proteins (Table 4). Secondly, compounds **4** and **11** have a good affinity for mTOR (−7.7 and −8.0 kcal·mol^−1^) and Bcl-2 (−8.1 and −7.9 kcal·mol^−1^), respectively. Therefore, the interaction with these proteins and their putative inhibitory effect may be related to the antiproliferative activity displayed by such quinone derivatives. Thirdly, the interactions are poor for CDC25A and B2M, with a maximum value of **−**5.9 kcal·mol**^−^**^1^ for compound **11** in CDC25A and a minimum value of −2.3 kcal·mol^−1^, respectively. 

The *Kd*, *LE* and Δ*EMW* values complemented these results. In some cases, compound **4** showed a better trend than compound **11** and vice versa. This depends on the environment in which the ligand is located; for example, compound **4** contains a phenyl group that promoted van der Waals (vdW) interactions such as π-stacking. In addition, the oxygen atom of the furyl group in compound **11** allows the formation of hydrogen bonds and the aromatic rings prioritize vdW interactions.

Figure 3 shows the interactions of compounds **4** and **11** with the amino acids corresponding to the mTOR, Bcl-2 and B2M proteins. It can be clearly observed that the interactions with the amino acids are not only due to the aromatic substituents, but also to the quinone rings, which are very reactive, thus allowing the formation of hydrogen bridges and π-π interactions. For complexes with lower affinity, interactions are weaker, and they are predominantly of the vdW type.

According to these data, compound **11** has similar interactions than compound **4** with regard to mTOR protein. Interestingly, both compounds interact with most of the amino acids identified as main targets of mTOR inhibitors such as Torin2, PP242 and PI-103 [39]. 

Although compounds **4** and **11** did not modify the expression levels of other genes (Table 3), their effects on CDC25A, TNF-α, HDAC4 and GSR proteins are shown in Figure 4. The contributing interactions with these proteins are the same as for the mTOR, Bcl-2 and B2M proteins (see above).

Altogether, the results show that mTOR appears to be a good intracellular target of quinones, explaining, at least in part, their inhibition of cancer cell proliferation. Actually, targeting the mTOR pathway has emerged as a promising therapeutic strategy for cancer treatment. To this end, several approaches have been developed hindering such mTOR contribution to cancer development including mTOR inhibitors [51,52,53,54,55], albeit with non-conclusive results. Indeed, despite the initial promise of mTOR inhibitors, resistance to these drugs is a major challenge in cancer treatment. In this context, several mechanisms have been proposed to explain resistance to mTOR inhibitors, including activation of compensatory signaling pathways, mutations in mTOR and its downstream effectors, and altered cellular metabolism. 

Dysregulation of mTOR pathway is a common feature of cancer cells, and targeting this pathway holds promise for cancer treatment. In this work, the implication of mTOR is underlined as a preliminary outcome but with high biological relevance. We have identified mTOR as a potential intracellular target using an in vitro approach; therefore, such a finding warrants further investigation. In this context, we are planning to perform additional experiments such as immunoprecipitation and immunoblotting procedures as well as an in vitro mTOR kinase assay. It will also be necessary to elucidate whether primary cells from cancer patients behave similarly to immortalized commercial cell lines. Future investigations should include a pre-clinical model to validate the interest of these new compounds as potential antitumoral drugs.

Yet, since other pathways may also be involved in the regulation of mTOR, they deserve a deeper assessment. Future studies should focus on whether compound **11** has a putative effect on the Nrf2 system of sensing environmental stress. In fact, a recent study has reported that Nrf2 regulates mTOR transcription [56]; therefore, it would be interesting to unveil the molecular link affecting mTOR, a key cellular protein in tumor metabolism.

## 3. Materials and Methods

### 3.1. Chemistry

#### 3.1.1. General Information 

All the solvents and reagents were purchased from different companies, such as Aldrich (St. Louis, MO, USA) and Merck (Darmstadt, Germany), and were used as supplied. Melting points (mp) were determined on a Stuart Scientific SMP3 (Staffordshire, UK) apparatus and are uncorrected. The IR spectra were recorded on an FT IR Bruker spectrophotometer, model Vector 22 (Bruker, Rheinstetten, Germany), using KBr disks, and the wave numbers are given in cm^−1^. ^1^H– and ^13^C–NMR spectra were recorded on a Bruker Ultrashield-300 instrument (Bruker, Ettlingen, Germany) in CDCl_3_ or DMSO-d_6_ at 300 and 75 MHz, respectively. Chemical shifts are expressed in ppm downfield relative to tetramethylsilane, and the coupling constants (*J*) are reported in Hertz. Data for the ^1^H–NMR spectra are reported as follows: s = singlet; br s = broad singlet; d = doublet; m = multiplet; and the coupling constants (*J*) are in Hz. Bi-dimensional NMR techniques and distortion-less enhancement by polarization transfer (DEPT) were used for the signal assignment. Chemical shifts are expressed in ppm downfield relative to tetramethylsilane, and the coupling constants (*J*) are reported in Hertz. The HRMS data for all final compounds were obtained using an LTQ-Orbitrap mass spectrometer (Thermo-Fisher Scientific, Waltham, MA, USA) with the analysis performed using an atmospheric-pressure chemical ionization (APCI) source, operated in positive mode. Silica gel Merck 60 (70–230 mesh, from Merck) was used for preparative column chromatography and thin layer chromatography (TLC) aluminum foil 60F_254_ was used for analytical thin layer chromatography.

#### 3.1.2. Synthesis of 2-Phenylamino-3-acyl-1,4-naphtoquinones **1**–**14**

The products required for the cytotoxic evaluation were synthesized according to our previously reported three-step procedure [9,29]. They included (a) solar photoacylation Friedel–Crafts reaction of 1,4-naphthoquinone (NQ) with aldehydes [57]; (b) oxidation of the resulting acylnaphthohydroquinones (2-acylNQ) with Ag_2_O to give the 2-phenylamino-1,4-naphtoquinones (AcylNQ); and (c) oxidative amination reaction of the products resulting in the previous step, with phenylamine, to produce the respective 2-phenylamino-3-acyl-1,4-naphtoquinones **1**–**14** (Figure 1).

The 2-AcNQH_2_ resulting in the step (a) were synthesized from 1,4-NQ and the following aldehydes: *n*-pentanal, *n*-nonanal, *n*-undecanal, benzaldehyde, 3-methoxybenzaldeyde, 4-methoxybenzaldeyde, 3,4-dimethoxybenzaldehyde, 3,4,5-trimethoxybenzaldehyde, 3-methoxy-4-hydroxybenzaldehyde, 4-methylbenzaldehyde, furan-2-carbaldehyde, thiophen-2-carbaldehyde, thiophen-3-carbaldehyde and pyrrole-2-carbaldehyde.

The structure of the known compounds **11** and **12** were confirmed based on their spectral data [9], and those of the remaining unknown analogues **1**–**10**, **13** and **14** were established by means of their IR, NMR and HRMS spectroscopy. The spectra of compounds were reported in Appendix A.

#### 3.1.3. General Procedure for the Preparation of 2-Phenylamino-3-acyl-1,4-naphtoquinones **1**–**10** and **13**–**14**

Suspensions of the acylnaphthohydroquinones (1.0 mmol), Ag_2_O (2.0 equiv.) and MgSO_4_ anhydrous (300 mg) in dichloromethane (30 mL) were left with stirring for 30 min at room temperature (rt). The mixtures were filtered, the solids were washed with dichloromethane (3 × 15 mL), and the filtrates containing the respective 2-acyl-1,4-naphthoquinones were evaporated under reduced pressure. The residues were dissolved in methanol (15 mL), the phenylamines (2 equiv.) and CeCl_3_·7H_2_O (5% mmol) were added to the solutions, and the mixtures were left, with stirring, at rt. The solvents were removed under reduced pressure, and the residues were column chromatographed over silica gel (petroleum ether/EtOAc) to yield the corresponding pure 2-phenylamino-3-acyl-1,4-naphtoquinones **1**–**10** and **13–14**.

*2-(Phenylamino)-3-hexanoylnaphthalene-1,4-dione***1**. (55%), red solid, mp: 124–126 °C. IR (KBr) ν_máx_ cm^−1^: 3431 (NH); 1687 (C=O); 1640 (C=O); 1595 (C=O). ^1^H–NMR (300 MHz, CDCl_3_) δ: 0.90 (t, 3H, *J* = 6.9 Hz, –COCH_2_–(CH_2_)_3_–CH_3_); 1.32 (m, 4H, –COCH_2_–CH_2_–CH_2_–CH_2_–CH_3_); 1.55 (m, 2H, –COCH_2_–CH_2_–(CH_2_)_2_–CH_3_); 3.04 (m, 2H, –COCH_2_–(CH_2_)_3_–CH_3_); 7.13 (m, 2H, H–arom); 7.29 (m, 1H, H–arom); 7.38 (m, 2H, H–arom); 7.65 (td, 1H, *J* = 7.5, 1.3 Hz, H–6 or H–7); 7.79 (td, 1H, *J* = 7.6, 1.4 Hz, H–7 or H–6); 7.93 (dd, 1H, *J* = 7.7, 0.9 MHz, H–5); 8.17 (dd, 1H, *J* = 7.8, 0.8 MHz, H–8); 12.09 (s, 1H, –NH). ^13^C–NMR (75 MHz, CDCl_3_) δ: 14.16; 22.73; 24.11; 31.65; 44.89; 112.78; 124.74 (2C); 126.25; 126.78; 126.98; 129.39 (2C); 131.04; 132.74; 133.53; 135.43; 139.20; 150.43; 181.69; 182.41; 205.39. HRMS (APCI): [M + H]^+^ calcd for C_22_H_21_NO_3_: 347.15214; found 347.15209.

*2-(Phenylamino)-3-decanoylnaphthalene-1,4-dione***2**. (55%), red solid, mp: 95–96 °C. IR (KBr) ν_máx_ cm^−1^: 3783 (NH); 1678 (C=O); 1638 (C=O); 1594 (C=O). ^1^H–NMR (300 MHz, CDCl_3_) δ: 0.88 (t, 3H, *J* = 6.6 Hz, –COCH_2_–(CH_2_)_7_–CH_3_); 1.29 (m, 12H, –COCH_2_–CH_2_–CH_2_–CH_2_–CH_2_–CH_2_–CH_2_–CH_2_–CH_3_); 1.54 (d, 2H, *J* = 6.8 Hz, –COCH_2_–CH_2_–(CH_2_)_6_–CH_3_); 3.04 (m, 2H, –COCH_2_–(CH_2_)_7_–CH_3_); 7.12 (d, 2H, *J* = 7.6 Hz, H–arom); 7.31 (d, 1H, *J* = 7.2 Hz, H–arom); 7.39 (t, 2H, *J* = 7.5 Hz, H–arom); 7.65 (td, 1H, *J* = 7.6, 1.1 Hz, H–6 or H–7); 7.80 (td, 1H, *J* = 7.6, 1.2 Hz, H–7 or H–6); 7.94 (d, 1H, *J* = 7.7 Hz, H–5); 8.17 (d, 1H, *J* = 7.8 Hz, H–8); 12.09 (s, 1H, –NH). ^13^C–NMR (75 MHz, CDCl_3_) δ: 14.15; 22.70; 24.31; 29.33; 29.36; 29.53; 29.58; 31.92; 44.83; 112.70; 124.63 (2C); 126.13; 126.67; 126.86; 129.27 (2C); 130.94; 132.62; 133.43; 135.31; 139.10; 150.31; 181.58; 182.30; 205.29. HRMS (APCI): [M + H]^+^ calcd for C_26_H_29_NO_3_: 403.21474; found 403.21159.

*2-(Phenylamino)-3-dodecanoylnaphthalene-1,4-dione***3**. (53%), red solid, mp: 100–101°C. IR (KBr) ν_máx_ cm^−1^: 3434 (NH); 1679 (C=O); 1638 (C=O); 1569 (C=O). ^1^H–NMR (300 MHz, CDCl_3_) δ: 0.87 (t, 3H, *J* = 6.3 Hz, –COCH_2_–(CH_2_)_9_–CH_3_); 1.28 (m, 16H, COCH_2_–CH_2_–CH_2_–CH_2_–CH_2_–CH_2_–CH_2_–CH_2_–CH_2_–CH_2_–CH_3_); 1.54 (m, 2H, COCH_2_–CH_2_–(CH_2_)_8_–CH_3_); 3.04 (t, 2H, *J* = 7.4 Hz, –COCH_2_–(CH_2_)_9_–CH_3_); 7.12 (d, 2H, *J* = 7.7 Hz, H–arom); 7.30 (d, 1H, *J* = 7.0 Hz, H–arom); 7.38 (t, 2H, *J* = 7.4 Hz, H–arom); 7.65 (t, 1H, *J* = 7.5 Hz H–6 or H–7); 7.79 (t, 1H, *J* = 7.6 Hz, H–7 or H–6); 7.93 (d, 1H, *J* = 7.6 Hz, H–5); 8.17 (d, 1H, *J* = 7.7 Hz, H–8); 12.09 (s, 1H, –NH). ^13^C–NMR (75 MHz, CDCl_3_) δ: 14.28; 22.83; 24.42; 29.49 (2C); 29.70 (2C); 29.79 (2C); 32.05; 44.95; 112.78; 124.73 (2C); 126.24; 126.78; 126.97; 129.38 (2C), 131.04; 132.73; 133.52; 135.43; 139.21; 150.42; 181.68; 182.41; 205.40. HRMS (APCI): [M + H]^+^ calcd for C_28_H_33_NO_3_: 431.21604; found 431.21814.

*2-(Phenylamino)-3-benzoylnaphthalene-1,4-dione***4**. (55%), orange solid, mp: 224–226 °C. IR (KBr) ν_máx_ cm^−1^: 3438 (NH); 1667 (C=O); 1592 (C=O); 1560 (C=O). ^1^H–NMR (300 MHz, CDCl_3_) δ: 6.85 (d, 2H, *J* = 7.0 Hz, H–arom); 7.00 (m, 3H, H–arom); 7.29 (m, 2H, H–arom); 7.46 (t, 1H, *J* = 7.4 Hz, H–arom); 7.55 (m, 2H, H–arom); 7.72 (td, 1H, *J* = 7.5, 1.3 Hz, H–7 or H–6); 7.80 (td, 1H, *J* =7.5, 1.3 Hz, H–6 or H–7); 7.90 (s, 1H, –NH), 8.12 (d, 1H, *J* = 7.6 Hz, H–5), 8.17 (d, 1H, *J* = 7.6 Hz, H–8). ^13^C–NMR (75 MHz, CDCl_3_) δ: 113.57; 126.19 (2xC); 126.69; 126.78; 127.18; 128.30 (2C); 128.93 (4C); 130.01; 132.85; 133.00; 133.17; 135.59; 136.85; 137.48; 143.77; 182.19; 182.38; 193.87. HRMS (APCI): [M + H]^+^ calcd for C_23_H_15_NO_3_: 353.10519; found 353.10196.

*2-(Phenylamino)-3-(3-methoxybenzoyl)naphthalene-1,4-dione***5**. (60%), orange solid, mp: 164–166 °C. IR (KBr) ν_máx_ cm^−1^: 3435 (NH); 1677 (C=O); 1652 (C=O); 1594 (C=O). ^1^H–NMR (300 MHz, CDCl_3_) δ: 3.74 (s, 3H, –OCH_3_); 6.86 (d, 2H, *J* = 6.8 Hz, H–arom); 7.01 (m, 5H, H–arom); 7.21 (d, 2H, *J* = 5.0 Hz, H–arom); 7.72 (t, 1H, *J* = 7.5 Hz, H–7 or H–6); 7.80 (t, 1H, *J* = 7.5 Hz, H–6 or H–7); 7.87 (s, 1H, –NH); 8.12 (d, 1H, *J* = 7.5 Hz, H–5), 8.17 (d, 1H, *J* = 7.6 Hz, H–8).^13^C–NMR (75 MHz, CDCl_3_) δ: 55.48; 112.00; 113.62; 120.13; 122.31; 126.30 (2C); 126.68; 126.79; 127.09; 128.95 (2C); 129.24; 129.98; 132.85; 132.98; 135.58; 136.83; 138.90; 143.63; 159.62; 182.14; 182.36; 193.66. HRMS (APCI): [M + H]^+^ calcd for C_24_H_17_NO_4_: 383.576; found 383.11242.

*2-(Phenylamino)-3-(4-methoxybenzoyl)naphthalene-1,4-dione***6**. (53%), orange solid, mp: 227–229 °C. IR (KBr) ν_máx_ cm^−1^: 3435 (NH); 1667 (C=O); 1659 (C=O); 1592 (C=O). ^1^H–NMR (300 MHz, CDCl_3_) δ: 3.83 (s, 3H, –OCH_3_); 6.77 (d, 2H, *J* = 8.7 Hz, H–arom); 6.86 (d, 2H, *J* = 7.0 Hz, H–arom); 7.00 (m, 3H, H–arom); 7.52 (d, 2H, *J* = 8.7 Hz, H–arom); 7.71 (t, 1H, *J* = 7.5 Hz, H–7 or H–6); 7.80 (m, 2H, –NH + H–6 or H–7); 8.11 (d, 1H, *J* = 7.6 Hz, H–5), 8.16 (d, 1H, *J* = 7.6 Hz, H–8). ^13^C–RMN (75 MHz, CDCl_3_) δ: 55.58; 113.52 (2C); 113.95; 126.16 (2C); 126.63; 126.76; 127.10; 128.78 (2C); 129.98; 130.97; 131.34 (2C); 132.78; 133.02; 135.52; 136.86; 143.45; 163.65; 182.21; 182.48; 192.20. HRMS (APCI): [M + H]^+^ calcd for C_24_H_17_NO_4_: 383.39608; found 383.39818.

*2-(Phenylamino)-3-(3,4-dimethoxybenzoyl)naphthalene-1,4-dione***7**. (63%), orange solid, mp: 217–219 °C. IR (KBr) ν_máx_ cm^−1^: 3435 (NH); 1675 (C=O); 1649 (C=O); 1618 (C=O). ^1^H–NMR (300 MHz, DMSO–d_6_) δ: 3.65 (s, 3H, –OCH_3_); 3.81 (s, 3H, –OCH_3_); 6.89 (m, 7H, H–arom); 7.30 (dd, 1H, *J* = 8.4, 1.5 Hz, H–arom); 7.87 (m, 3H, H–5 + H–6 + H–7); 8.11 (d, 1H, *J* = 7.5 Hz, H–8); 9.33 (s, 1H, –NH). ^13^C–NMR (75 MHz, DMSO–d_6_) δ: 55.38; 55.72; 109.88; 110.33; 113.37; 124.28; 125.53; 125.75; 126.07; 126.27 (2C); 127.91 (2C); 130.31; 130.72; 132.61; 132.80; 135.07; 137.92; 144.66; 148.24; 152.94; 181.53; 182.10; 192.02. HRMS (APCI): [M + H]^+^ calcd for C_25_H_19_NO_5_: 413.12632; found 413.12275.

*2-(Phenylamino)-3-(3,4,5-trimethoxybenzoyl)naphthalene-1,4-dione***8**. (55%), orange solid, mp: 209–210° C. IR (KBr) ν_máx_ cm^−1^: 3435 (NH); 1683 (C=O); 1657 (C=O); 1509 (C=O). ^1^H–NMR (300 MHz, DMSO–d_6_) δ: 3.69 (s, 6H, –OCH_3_); 3.72 (s, 3H, –OCH_3_); 6.74 (s, 2H, H–arom); 6.82 (m, 2H, H–arom); 6.95 (m, 3H, H–arom); 7.85 (m, 2H, H–5 + H–7 or H–6); 7.96 (d, 1H, *J* = 7.3 Hz, H–6 or H–7); 8.12 (d, 1H, *J* = 6.7 Hz, H–8); 9.35 (s, 1H, –NH). ^13^C–NMR (75 MHz, DMSO–d_6_) δ: 56.10 (2C); 60.24; 106.24; 112.84; 125.55; 125.71; 126.05; 126.24 (2C); 128.00 (3C); 130.44; 132.75 (2C); 132.78; 135.01; 137.98; 141.94; 144.99; 152.43 (2xC); 181.53; 182.09; 192.43. HRMS (APCI): [M + H]^+^ calcd for C_26_H_21_NO_6_: 443.13689; found 443.13299.

*2-(Phenylamino)-3-(4-hydroxy-3-methoxybenzoyl)naphthalene-1,4-dione***9**. (54%), orange solid, mp: 194–195 °C. IR (KBr) ν_máx_ cm^−1^: 3433 (NH); 1679 (C=O); 1565 (C=O); 1503 (C=O). ^1^H–NMR (300 MHz, CDCl_3_) δ: 3.78 (s, 3H, –OCH_3_); 6.15 (s, 1H, –OH); 6.79 (d, 1H, *J* = 8.2 Hz, H–arom); 6.85 (m, 2H, H–arom); 6.94 (d, 1H, *J* = 1.7 Hz, H–arom); 7.01 (m, 3H, H–arom); 7.23 (dd, 1H, *J* = 8.2, 1.8 Hz, H–arom); 7.71 (dt, 1H, *J* = 7.5, 3.8 Hz, H–7 or H–6); 7.79 (m, 2H, –NH + H–6 or H–7); 8.12 (d, 1H, *J* = 7.6 Hz, H–5); 8.16 (d, 1H, *J* = 7.7 Hz, H–8). ^13^C–NMR (75 MHz, CDCl_3_) δ: 56.09; 109.54; 113.61; 113.83; 125.55; 126.40 (2C); 126.66; 126.80; 126.92; 128.78 (2C); 129.95; 130.89; 132.82; 133.00; 135.55; 136.82; 143.34; 146.57; 150.63; 182.19; 182.43; 192.28. HRMS (APCI): [M + H]^+^ calcd for C_24_H_17_NO_5_: 399.11067; found 399.11316.

*2-(Phenylamino)-3-(4-methylbenzoyl)naphthalene-1,4-dione***10**. (50%), red solid, mp: 224–226 °C. IR (KBr) ν_máx_ cm^−1^: 3434 (NH); 1679 (C=O); 1658 (C=O); 1604 (C=O). ^1^H-NMR (300 MHz, CDCl_3_) δ: 2.36 (s, 3H, –CH_3_); 6.86 (d, 2H, *J* = 7.3 Hz, H–arom); 7.00 (m, 3H, H–arom); 7.09 (d, 2H, *J* = 7.9 Hz, H–arom); 7.46 (d, 2H, *J* = 8.0 Hz, H–arom); 7.71 (t, 1H, *J* = 7.5 Hz, H–7 or H–6); 7.79 (t, 1H, *J* = 7.5 Hz, H–6 or H–7); 7.89 (s, 1H, -NH); 8.11 (d, 1H, *J* = 7.6 Hz, H–5); 8.16 (d, 1H, *J* = 7.5 Hz, H–8). ^13^C-NMR (75 MHz, CDCl_3_) δ: 21.87; 113.78; 126.06 (2xC); 126.64; 126.76; 127.09; 128.87 (2xC); 129.03 (2xC); 129.09 (2xC); 130.00; 132.79; 133.01; 135.21; 135.53; 136.91; 143.61; 144.03; 182.23; 182.44; 193.46. HRMS (APCI): [M + H]^+^ calcd for C_24_H_17_NO_3_: 367.12084; found 367.12371.

*2-(Phenylamino)-3-(thiophene-3-carbonyl)naphthalene-1,4-dione***13**. (50%), orange solid. mp: 187–189 °C. IR (KBr) ν_máx_ cm^−1^: 3432 (NH); 1677 (C=O); 1657 (C=O); 1561 (C=O). ^1^H–NMR (300 MHz, CDCl_3_) δ: 6.87 (m, 2H, H–arom); 7.06 (m, 4H, H–arom); 7.12 (m, 1H, H–arom); 7.71 (m, 2H, H–7 or H–6 + H–arom); 7.80 (t, 1H, *J* = 7.5 Hz, H–6 or H–7); 7.86 (s, 1H, –NH); 8.14 (t, 2H, *J* = 8.3 Hz, H–5 + H–8). ^13^C-NMR (75 MHz, CDCl_3_) δ: 114.40; 125.83; 125.96 (2C); 126.67; 126.79; 127.09; 127.16; 128.87 (2C); 129.91; 132.86; 132.95; 133.62; 135.62; 136.81; 143.27; 143.30; 181.94; 182.50; 187.07. HRMS (APCI): [M + H]^+^ calcd for C_21_H_13_NO_3_S: 359.06161; found 359.05989

*2-(Phenylamino)-3-(1H-pyrrole-2-carbonyl)naphthalene-1,4-dione***14**. (55%), red solid, mp: 210–212 °C. IR (KBr) ν_máx_ cm^−1^: 3439 (NH); 1670 (C=O); 1615 (C=O); 1591 (C=O). ^1^H–NMR (300 MHz, CDCl_3_) δ: 6.19 (m, 1H, H–arom); 6.69 (s, 1H, H–arom); 6.89 (d, 3H, *J* = 7.6 Hz, H–arom); 7.04 (d, 3H, *J* = 6.8 Hz, H–arom); 7.71 (t, 1H, *J* = 7.5 Hz, H–7 or H–6); 7.80 (m, 2H, –NH + H–6 or H–7); 8.15 (d, 2H, *J* = 8.2 Hz, H–8 + H–5); 8.97 (s, 1H, –NH). ^13^C–NMR (75 MHz, CDCl_3_) δ: 110.94; 113.70; 118.83; 125.29; 125.80 (2C); 126.62; 126.84 (2C); 128.41 (2C); 129.96; 132.75; 133.08; 133.60; 135.52; 136.81; 143.27; 181.52; 181.89; 182.59. HRMS (APCI): [M + H]^+^ calcd for C_21_H_14_N_2_O_3_: 342.10044; found 342.099234.

#### 3.1.4. Molecular Descriptors

Calculation of lipophilicity (ClogP) and molar refractivity (CMR) was assessed by using the ChemBioDraw Ultra 11.0 software and the obtained values are shown in Table 3. Redox potentials of 2-phenylamino-3-acyl-1,4-naphtoquinones **1**–**14** were measured by cyclic voltammetry at room temperature in acetonitrile as solvent using a platinum electrode and 0.1M tetraethylammonium tetrafluoroborate as the supporting electrolyte [58].

### 3.2. Cytotoxic Assays

#### 3.2.1. Cell Lines and Cell Cultures

Human cancer cell lines from bladder (T24), prostate (DU–145), breast (MCF–7) and non-tumor HEK-293 cells were obtained from the American Type Culture Collection (ATCC, Manassas, VA, USA). The cultures were maintained at a density of 1–2 × 10^5^ cells/mL and the medium was changed at 48- and 72-h intervals. They were cultured in high-glucose Dulbecco’s modified Eagle medium (Gibco, Grand Island, NY, USA) supplemented with 10% fetal calf serum, penicillin (100 U/mL) and streptomycin (100 μg/mL). All cultures were kept at 37 °C in 95% air/5% CO_2_ at 100% humidity. Phosphate-buffered saline (PBS) was purchased from Gibco. Cells were incubated at the indicated times at 37 °C with or without compounds **1**–**14** at various concentrations.

#### 3.2.2. Cell Survival Assays

The cytotoxicity of the compounds **1**–**14** was assessed by following the reduction of MTT (3-(4,5-Dimethylthiazol-2-yl)-2,5-diphenyltetrazolium bromide) to formazan blue [59]. Cells were seeded into 96-well plates at a density of 10,000 cells/well for 24 h and then they were further incubated for 24 h with or without the quinones. Doxorubicin was used as a standard chemotherapeutic agent (positive control). Cells were washed twice with warm PBS and further incubated with MTT (0.5 mg/mL) for 2 h at 37 °C. Blue formazan crystals were solubilized by adding 100 µL DMSO/well, and the optical density of colored solutions was subsequently read at 550 nm in a microplate reader Tecan infinite M200 Pro (Männedorf, Switzerland). The compounds **1**–**14** were dissolved in DMSO (stock solution at 100 mM) and further diluted to be evaluated at the following concentrations: 0 μM, 1 μM, 10 μM, 20 μM, 40 μM, 60 μM, 80 μM and 100 μM. Results are expressed as % of MTT reduction compared to untreated control conditions. The IC_50_ values were calculated using the GraphPad Prism 8.0.2 software (San Diego, CA, USA).

### 3.3. Quantitative Real-Time PCR (qPCR) Assay

The DU-124 cells were cultured as previously mentioned. They were seeded into 6-well plates (2 × 10^5^ cells/well) and, after 24 h of incubation, they were treated for 48 h with compounds **4** and **11** (at 32 and 68 μM, respectively). Afterwards, they were washed with phosphate-buffered saline. The cellular lysate was prepared with E.Z.N.A.^®^RNA-Lock Reagent (Omega Bio-tek, Norcross, GA, USA) to preserve and immediately stabilize the total RNA for the subsequent gene expression assays. The total RNA isolated from the cells using the E.Z.N.A.^®^HP Total RNA Isolation Kit (Omega Bio-tek) was reverse-transcribed to cDNA using the AffinityScript QPCR cDNA Synthesis Kit (Agilent Technologies, Santa Clara, CA, USA) and 1000 ng of the RNA sample.

The cDNA synthesized was employed for qPCR using Brilliant III Ultra-Fast SYBR^®^Green QPCR Master Mix (Agilent Technologies) in an Mx3000P qPCR System (Agilent Technologies), employing a 96-well plate with 20 μL of PCR reaction per well and 10 pmol each of forward and reverse gene-specific primers. Nine genes were analyzed: *B2M, Bcl2, CDC25A, CCN2, GSR, HDAC4, mTOR, TNF, TP53* and their quantitative real-time (qPCR) primer sequences are reported in Appendix A. The relative gene expressions were determined using Beta-2-microglobulin (*B2M*) as housekeeping, and the delta-delta Ct method (2^−∆∆Ct^ method) with regard to the vehicle-treated group (i.e., the reference group). Five biological replicates were used from each group (treated and reference group). The qPCR reactions were run by duplicates and negative controls contained no cDNA, as previously reported [60]. The GraphPad Prism 8.0.2 software was used for statistical analyses of the relative gene expressions. The comparisons between means were performed using one-way analysis of variance (ANOVA) and Dunnett’s multiple comparisons test. All statistical analyses were performed with a significance level of *p* < 0.05.

### 3.4. In Silico Studies

#### 3.4.1. Molecular Docking

The compounds **4** and **11** were docked as potential inhibitors of the following proteins: Human CDC25A [61], TNF-α [62], Human HDAC4 [63], human glutathione reductase GSR [64], mTOR kinase [65], beta2-microglobulin B2M fibril [66] and human Bcl-2 promoter [67], using AutoDock Vina (v 1.0.2). The three-dimensional coordinates of all structures were optimized using MOPAC2016 software by PM6-D3H4 semi-empirical method [68,69]. The ligand files were prepared using the AutoDockTools package [70]. The crystal structure of CDC25A (PDB Code: 1C25), TNF-α (PDB Code: 2AZ5), HDAC4 (PDB Code: 2VQJ), GSR (PDB Code: 3DK9), mTOR (PDB Code: 4JSN), B2M (PDB Code: 6GK3) and Bcl-2 (PDB Code: 2W3L), were downloaded from the Protein Data Bank [71]. The CDC25A, TNF-α, HDAC4, GSR, mTOR, B2M and Bcl-2 were treated with the Schrödinger’s Protein Preparation Wizard [72]; polar hydrogen atoms were added, nonpolar hydrogen atoms were merged, and charges were assigned. Docking was treated as rigid and carried out using the empirical free energy function and the Lamarckian Genetic Algorithm provided by AutoDock Vina [73,74,75]. The grid map dimensions were 20 × 20 × 20 Å^3^. The centre of the binding site were the following coordinates for each of the proteins studied (Table 6). Each binding site coordinate shown in Table 6 represents the position obtained from the literature of ligands from the same aromatic chemical class and the geometric center of each co-crystallized ligand with the protein.

All other parameters were set as the default defined by AutoDock Vina. Dockings were repeated 20 times with space search exhaustiveness set to 100. The best interaction binding energy (kcal·mol^−1^) was selected for evaluation. Docking results 3D representations were used Discovery Studio 3.1 (Accelrys, San Diego, CA, USA) molecular graphics system was used.

#### 3.4.2. Ligand Efficiency

Ligand efficiency (LE) calculations were performed using the *K_d_* parameter. The latter corresponds to the dissociation constant between a ligand/protein and its value indicates the bond strength between the ligand/protein [101,102]. Low values indicate strong binding of the molecule to the protein. *K_d_* calculations were conducted using Equations (1) and (2):(1)$$∆G^{0}=-2.303RT\log\left(K_{d}\right)$$
(2)$$K_{d}=10^{\frac{∆G^{0}}{2.303\mathrm{RT}}},$$
where ∆*G*^0^ is the binding energy (kcal·mol^−1^) obtained from docking experiments, *R* is the gas constant and *T* is the temperature in Kelvin in standard conditions of aqueous solution at 298.15 K, neutral pH and remaining concentrations of 1 M. The *LE* Equation (3) allows us to compare molecules according to their average binding energy [103], and is computed as the ratio of binding energy per non-hydrogen atom [101,102,104]:(3)$$LE=-\frac{2.303RT}{HAC}\log\left(K_{d}\right),$$
where *K_d_* is obtained from Equation (2) and HAC denotes the heavy atom count (i.e., number of non-hydrogen atoms) in a ligand.

To complement this ligand efficiency study, an additional analysis of the size of the molecules in relation to the binding energy was implemented. Score Normalization Based on the Number of Non-Hydrogen Atoms—this score-based approach ($IE_{norm,\;\mathrm{binding}}$) is biased towards the selection of high molecular weight compounds because of the contribution of the compound size to the energy score [105]. Such biasing behavior was observed to depend on the shape and chemical properties of the binding pocket. The procedure starts with the normalization of the binding energy ($IE_{\mathrm{binding}\;}$) by the number of heavy atoms (HAC) or by a selected power of HAC in each respective compound. This normalization approach shifts the MW distribution of selected compounds. In the present study, the following Equation (4) was used to calculate the normalized binding energy value.
(4)$$IE_{norm,\;\mathrm{binding}\;}=\frac{IE_{\mathrm{binding}\;}}{{\mathrm{HAC}}^{\frac{1}{2}}}$$

An important aspect of normalizing binding energy is the ability to bias selection towards lower MW compounds, thereby identifying compounds more appropriate for lead optimization. Importantly, ligand-based post-docking structural clustering leads to the selection of diverse compounds, many of which would have been lost through selection based on binding energy alone. Therefore, it is important to establish a relationship between binding energy and MW of **4** and **11** compounds.

### 3.5. Physicochemical, Pharmacokinetic, and Drug Likeness Properties

SwissADME (http://swissadme.ch, accessed on 8 August 2022) [106] and pkCSM online tools (http://biosig.unimelb.edu.au/pkcsm/prediction, accessed on 15 August 2022) [107] were utilized to predict physicochemical, pharmacokinetic (ADMET) and drug likeness properties of compounds **4** and **11**.

### 3.6. Statistical Analysis

GraphPad Prism 8.0.2 software (San Diego, CA, USA) was used for statistical analysis. The IC_50_ value (concentration of compounds causing half-maximal responses) was established by regression analysis.

## 4. Conclusions

This study demonstrates that a new set of 2-phenylamino-3-acyl-1,4-naphthoquinones, prepared through an environmentally friendly protocol, were evaluated against three human cell lines (DU-145, MCF-7, and T24). Compounds **4** and **11** appeared as the most active compound against proliferation of DU-145 human cancer cells. Based on previous elements already discussed, we would like to suggest that compound **11** is a potential suitable molecule that deserves to be further developed. In addition, its lipophilia allows it to traverse cell membranes to exert its cytotoxic action. We propose that mTOR is an interesting intracellular target and its dysregulation by compound **11** may affect cancer cells growth.

## Data Availability

Not applicable.

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
