# Peer review of "Discovery of New 2-Phenylamino-3-acyl-1,4-naphthoquinones as Inhibitors of Cancer Cells Proliferation: Searching for Intra-Cellular Targets Playing a Role in Cancer Cells Survival"

_molecules, 2023, doi:10.3390/molecules28114323_

Round 1
Reviewer 1 Report (New Reviewer)
Paper reports synthesis, in vitro biological evaluations and molecular docking simulation of a series of 2-phenylamino-3-acyl-1,4-naphtoquinones.
Collectively this paper is well written and well organized and only minor revisions are necessary:
1) please add a supporting material file with images of NMR spectra of new synthesize compounds;
2) Paragraph 2 (“results and discussion”) should be divided in sub-session as anti-proliferative evaluation section, physochemical, PK and DMPK evaluation section, mRNa expression evaluation section, molecular docking section;
3) in table 4 Brenk filter evaluation calculated with SwissADME should be added;
4) please insert in all text number of compounds 4 and 11 in bold;
5) experimental part: please verify if NMR spectra are calculated with 300MHz instrument or 400 MHz instruments (line 336); please check:
6) Line 325: please insert space after “transcription”;
7) line 512: please insert space after “blue”.
Author Response
Referee n°1
Paper reports synthesis, in vitro biological evaluations and molecular docking simulation of a series of 2-phenylamino-3-acyl-1,4-naphtoquinones.
Collectively this paper is well written and well organized and only minor revisions are necessary:
We thank the referee for her/his positive and constructive opinion with regard to our work.
1) please add a supporting material file with images of NMR spectra of new synthesize compounds;
Answer: the supporting material file with images of NMR spectra of new synthesize compounds is included in the Supplementary Material enclosed in this resubmission process.
2) Paragraph 2 (“results and discussion”) should be divided in sub-session as anti-proliferative evaluation section, physicochemical, PK and DMPK evaluation section, mRNA expression evaluation section, molecular docking section;
Answer: we thank the comment made by the referee. In the revised manuscript, the section Results and Discussion, were divided according to her/his suggestion.
3) in table 4 Brenk filter evaluation calculated with SwissADME should be added;
Answer: according to referee’s suggestion, the Brenk filter evaluation was included in table 4.
4) please insert in all text number of compounds 4 and 11 in bold;
Answer: it was done.
5) experimental part: please verify if NMR spectra are calculated with 300MHz instrument or 400 MHz instruments (line 336); please check:
Answer: it was revised and corrected
6) Line 325: please insert space after “transcription”;
Answer: it was done.
7) line 512: please insert space after “blue”.
Answer: it was done.
Reviewer 2 Report (New Reviewer)
Suggestion: minor revisions
I commend the authors for providing extensively researched article for their work. Here are some minor suggestions to further enhance the manuscript.
Suggestions:
1. The version I downloaded had a few lines in the introduction highlighted. Is there a reason why it is highlighted in the main text?
2. In Table 2 – could you explain the unit of the numbers below each of the cell lines? Is it all IC50 – so the values are in mg concentration of the drug? Or is it % of cell survival? Currently it’s a bit unclear. May be sentence before the table, or a line in the table heading would make it helpful for the readers to better understand.
3. For table 5 – will it be possible to provide a graph for each of the genes and compare the level of expression between compound 4 and 11? It would be interesting to observe visually the differences. Also there is an * for compound 11 on the expression of mTOR but that is not explained in the table heading. Is that statistically significant?
4. I would recommend replacing ‘a good molecule’ with ‘a potential suitable molecule’ in conclusion and anywhere else
5. Please add a paragraph each on limitations (example - not tested on primary cells from cancer patients) and future directions to your conclusions.
Minor editing of English language required
Author Response
Referee n°2
I commend the authors for providing extensively researched article for their work. Here are some minor suggestions to further enhance the manuscript.
We thank the referee for her/his positive and constructive opinion regarding our manuscript
- The version I downloaded had a few lines in the introduction highlighted. Is there a reason why it is highlighted in the main text?
Answer: we apologize for such mistake. Indeed, it correspond to old corrections that we forgot to erase. This is now corrected in the revised manuscript.
- In Table 2 – could you explain the unit of the numbers below each of the cell lines? Is it all IC50– so the values are in mg concentration of the drug? Or is it % of cell survival? Currently it’s a bit unclear. May be sentence before the table, or a line in the table heading would make it helpful for the readers to better understand.
Answer: we apologize for such misleading, but as the referee can verify in the former manuscript, it was indicated just at the beginning of the title in Table 2. Indeed, it is mentioned that IC50 values are expressed in mM.
- For table 5 – will it be possible to provide a graph for each of the genes and compare the level of expression between compound 4 and 11? It would be interesting to observe visually the differences. Also there is an * for compound 11 on the expression of mTOR but that is not explained in the table heading. Is that statistically significant?
Answer: Once again, we apologize to induce such a confusing situation. Indeed, the value for mTOR was indicated with an asterisk (*) that means a statistically significant difference. It was indicated in the last sentence in the legend of the Table.
- I would recommend replacing ‘a good molecule’ with ‘a potential suitable molecule’ in conclusion and anywhere else
Answer: it was done
- Please add a paragraph each on limitations (example - not tested on primary cells from cancer patients) and future directions to your conclusions.
Answer: According to the referee’s suggestion, we have included some sentences (lines 342-351) with regard to limitations of the study (only in vitro assays) and how are we dealing to go further with our research project.
Comments on the Quality of English Language: Minor editing of English language required
The English language was carefully revised and improved
This manuscript is a resubmission of an earlier submission. The following is a list of the peer review reports and author responses from that submission.
Round 1
Reviewer 1 Report
Authors demonstrated that compound 11 decreased the expression of mTOR expression. But, the relative expression levels are only 0.72, suggesting that the effect is not large. I do not agree that this slightly decreased expression causes significant effect. In addition, authors do not show the mechanism how compound 11 decreased mTOR expression. This is critical.
Authors suggest the interaction of the compound and some proteins. To prove that, immunoprecipitation assay (ip+ib) is needed.
I do not think that further English editing is needed.
Author Response
Referee #1
Authors demonstrated that compound 11 decreased the expression of mTOR expression. But, the relative expression levels are only 0.72, suggesting that the effect is not large. I do not agree that this slightly decreased expression causes significant effect. In addition, authors do not show the mechanism how compound 11 decreased mTOR expression. This is critical.
Answer: We agree that mTOR expression is slightly decreased but we have already observed that the same kind of quinones can inhibit Akt (reference quoted #27). These two evidences led to the hypothesis that the PI3K/Akt/mTOR signaling pathway is impaired. Consequently, we suggest that mTOR may be involved and warrant further investigation. Elucidation of the molecular mechanism by which a given compound exerts its action is a major issue that requires methods that are more sophisticated. It should be remind that such results are preliminary as it was highlighted in the original manuscript. Indeed, we explained the limitations of the study and identified future research directions based on the conclusions.
Authors suggest the interaction of the compound and some proteins. To prove that, immunoprecipitation assay (ip+ib) is needed.
Answer: We thanks the suggestion made by the referee. Immunoprecipitation is an appropriate assay to determine whether the link between the protein and compound 11 is strong enough to remain stable. However, it may be possible that such an interaction does not involve a physical and stable link, but may be sufficient to modify protein activity. Since the aim of this study was to investigate the antiproliferative activity of quinones, and (if possible) to identify putative targets, we are planning to conduct western blots as well as an in vitro mTOR kinase assay. In this context, we thank the referee for the suggestion, and the immunoprecipitation assay will also be performed in the next article.
Reviewer 2 Report
In the present manuscript, the authors evaluated the in vitro cytotoxicity of 14 novel naphthoquinones and selected compounds 4 and 11 for exploring their effects on several genes that are representative of several pathways involved in carcinogenesis and cancer cell survival. However, this study is at a very low level, and there is a serious lack of experimental data to support its conclusions.
None.
Author Response
Referee #2
In the present manuscript, the authors evaluated the in vitro cytotoxicity of 14 novel naphthoquinones and selected compounds 4 and 11 for exploring their effects on several genes that are representative of several pathways involved in carcinogenesis and cancer cell survival. However, this study is at a very low level, and there is a serious lack of experimental data to support its conclusions.
Answer: We thanks the referee for her/his comments. We partially agree with such criticism (“low level”) because we believe that the two main findings in this study (biological activity and intracellular target) should be discussed separately. To our understanding, the experimental data leading to the identification of compounds 4 and 11 as molecules bearing an antiproliferative effect are well documented, and the experimental approach is sound. This work is in line with our previous reports on the identification of both potential bioactive compounds and cell targets of interest.
Regarding the intracellular target, we agree that such a conclusion is a preliminary fact that should be further explored. However, the suggestion we made was based on various simultaneous approaches, such as physical-chemistry analysis, molecular docking, and ADME study. Nevertheless, the potential role of mTOR is a preliminary conclusion. In the original manuscript, we explained that this point represents a limitation of this study. Current studies are in progress using additional assays such as western blot and in vitro kinase assays, among other methods.
Reviewer 3 Report
COMMENTS
Overall, this is a well written paper with an interesting result on cancer research. The aim of the research was to investigate whether the in vitro cytotoxicity of a novel 80 designed series of 2-phenylamino-3-acyl-1,4-napththoquinones may be associated with an 81 altered expression of some representative genes of major cellular pathways playing a role 82 in carcinogenesis and cancer cells survival.
The results are based on rational working hypothesis, well described and with a correct research design. Congratulations to the authors for the nice work conducted
INTRODUCTION
The introduction must provide sufficient background information for readers to understand the research aim, however the authors should clarify the importance of this topic and the actual knowledge in this area. It seems like some important clarification is missing.
Motivations for this study are more than clear and the objectives are clearly defined at the Introduction, the argumentation in this part was concise.
METHODS
The methodology proposed to reach the aim of the study look appropriate, well designed and conducted.
RESULTS
Results paragraphs include the most relevant data.
All of the tables include specific and well presented statistic.
The figures allow to a better comprehension of the manuscript.
DISCUSSION
All possible interpretations of the data considered are consistent, support the research aim and allow to a better understanding.
Explain limitation of the study and future research line according to the study conclusion
LITERATURE CITED
The literature cited is relevant to the study.
Author Response
Referee #3
Overall, this is a well written paper with an interesting result on cancer research. The aim of the research was to investigate whether the in vitro cytotoxicity of a novel designed series of 2-phenylamino-3-acyl-1,4-napththoquinones may be associated with an altered expression of some representative genes of major cellular pathways playing a role 82 in carcinogenesis and cancer cells survival. The results are based on rational working hypothesis, well described and with a correct research design. Congratulations to the authors for the nice work conducted
We thanks the referee for her/his stimulating comment.
INTRODUCTION: The introduction must provide sufficient background information for readers to understand the research aim, however the authors should clarify the importance of this topic and the actual knowledge in this area. It seems like some important clarification is missing. Motivations for this study are more than clear and the objectives are clearly defined at the Introduction, the argumentation in this part was concise.
METHODS: The methodology proposed to reach the aim of the study look appropriate, well designed and conducted.
RESULTS: Results paragraphs include the most relevant data. All of the tables include specific and well presented statistic. The figures allow to a better comprehension of the manuscript.
DISCUSSION: All possible interpretations of the data considered are consistent, support the research aim and allow to a better understanding. Explain limitation of the study and future research line according to the study conclusion.
LITERATURE CITED: The literature cited is relevant to the study.
Answer: We appreciate the comments of the referee. According to her/his suggestion, some sentences have been included in the introduction section of the revised manuscript. Regarding other comments, we thank the referee for her/his kind words, pointing out the real value of both the experimental approach and the results we obtained. We are also grateful for the comments made by the referee and more specifically, for the point she/he made about the preliminary conclusions regarding mTOR. Finally, we express our gratitude for the referee’s constructive comments.
Round 2
Reviewer 1 Report
I do not agree with author’s opinions, and the evidence of impaired PI3K/Akt/mTOR signal is very week. Authors did not show possible mechanism about the action of the compound. Although authors explained limitation, they should show stronger evidence and should not show just preliminary results.
Authors also did not conduct immunoprecipitation assay that I suggested.
Author Response
I do not agree with author’s opinions, and the evidence of impaired PI3K/Akt/mTOR signal is very week.
Answer: We agree that it is weak but statistically significant. In addition, three lines of evidences, pointing out mTOR protein as potential target of the bioactive quinones, have been shown. They included previous work showing that quinones decreased Akt levels, reduced mTOR expression and had good interactions with mTOR protein.
Authors did not show possible mechanism about the action of the compound. Although authors explained limitation, they should show stronger evidence and should not show just preliminary results
Answer: What we try to explain is that deciphering a mechanism of action is not a trivial and an easy job. If the referee is asking for some hypothesis, obviously we can develop some ideas. Indeed, the impairment of mTOR activity can occur due to various reasons, including genetic mutations, environmental factors, and changes in nutrient availability. Under our experimental conditions, mutations and nutrient availability can be discarded. Regarding other factors evoked as molecular explanations for this impairment, some common mechanisms include oxidative stress, dysregulated signaling pathways and drug inhibition. Indeed, it is known that ROS can impair mTOR activity by inhibiting the mTORC1 pathway. ROS can directly inhibit mTORC1 by oxidizing its components, or indirectly by activating other signaling pathways that inhibit mTORC1. This explanation is however, complex to develop because we did not find differences in the quinone redox potentials while they display different antiproliferative activities. Dysregulation of signaling pathways, for instance decreased activation of the PI3K/Akt/mTOR, or increased activation of the AMPK/mTOR, can impair mTOR activity. We have some experimental evidences suggesting that the former signaling pathway may be inhibited. Therefore, the second pathway should be examined. Finally, rapamycin and its analogs inhibit mTOR activity by binding to its regulatory protein, FKBP12. The rapamycin-FKBP12 complex then binds to a specific site on mTOR that is located within the mTOR kinase domain. This binding induces a conformational change in mTOR and it prevents mTOR from interacting with its downstream targets and inhibits its activity. We have made preliminary molecular docking studies by comparing the interactions between FKBP12 with either compound 11 or rapamycin. The values obtained showed that interactions of compound 11 are slightly weaker than rapamacyn.
If the referee agree, we can include all this part in the discussion section.
Authors also did not conduct immunoprecipitation assay that I suggested.
Answer: we have already explained that we will set up the experimental conditions to perform such an experiment. Meanwhile, we need to buy antibodies and other reagents, for which we need money, and consequently to get it, we need to apply for new funds.
Reviewer 2 Report
The manuscript has shown some improvement, but it is still not up to the standards required for publication in Molecules. The abstract is well-written, but lacks sufficient experimental data to support its claims. For example, there is no data demonstrating the ability of compound 11 to inhibit mTOR activity or reduce mTOR expression.
Author Response
The manuscript has shown some improvement, but it is still not up to the standards required for publication in Molecules. The abstract is well-written, but lacks sufficient experimental data to support its claims. For example, there is no data demonstrating the ability of compound 11 to inhibit mTOR activity or reduce mTOR expression
Answer: We disagree with this comment because the paper is not focused on mTOR, and we never claimed it !!. Yet, while this part is preliminary, three lines of evidences are pointing out mTOR protein as potential target of the bioactive quinones. They included previous work showing that quinones decreased Akt levels, reduced mTOR expression and had good interactions with mTOR protein. Therefore, we cannot accept the sentence that no experimental data is supporting our work. Such a criticism ignores the entire work performed. The results presented in this study are strong enough, they are backed by a suitable and sound experimental approach and they are in agreement with the original aim that is whether such original quinones display an antiproliferative activity. Once again, we want to insist that mTOR was not the central subject of our investigation.